# Increased mTOR Signaling and Impaired Autophagic Flux Are Hallmarks of SARS-CoV-2 Infection

Érika Pereira Zambalde [1,†], Thomaz Luscher Dias [2,3,†], Grazielle Celeste Maktura [4], Mariene R. Amorim [4], Bianca Brenha [4], Luana Nunes Santos [4], Lucas Buscaratti [4], João Gabriel de Angeli Elston [4], Mariana Camargo Silva Mancini [1], Isadora Carolina Betim Pavan [1,5], Daniel A. Toledo-Teixeira [6], Karina Bispo-dos-Santos [6], Pierina L. Parise [6], Ana Paula Morelli [1], Luiz Guilherme Salvino da Silva [1], Ícaro Maia Santos de Castro [2], Tatiana D. Saccon [7,8,9], Marcelo A. Mori [7,8,9], Fabiana Granja [6,10], Helder I. Nakaya [2,11], Jose Luiz Proenca-Modena [6,7], Henrique Marques-Souza [4,*] and Fernando Moreira Simabuco [1,12,*]

1   Multidisciplinary Laboratory of Food and Health (LabMAS), School of Applied Sciences (FCA), University of Campinas (UNICAMP), Limeira 13484-350, SP, Brazil
2   Computational Systems Biology Lab (CSBL), Departamento de Análises Clínicas e Toxicológicas, Faculdade de Ciências Farmacêuticas, Universidade de São Paulo, São Paulo 05508-000, SP, Brazil
3   Laboratório de Genética Bioquímica, Departamento de Bioquímica e Imunologia, Instituto de Ciências Biológicas, Universidade Federal de Minas Gerais (UFMG), Belo Horizonte 31270-901, MG, Brazil
4   Brazilian Laboratory on Silencing Technologies (BLaST), Department of Biochemistry and Tissue Biology, Institute of Biology, University of Campinas, Campinas 13083-872, SP, Brazil
5   Laboratory of Signaling Mechanisms (LMS), School of Pharmaceutical Sciences (FCF), University of Campinas (UNICAMP), Campinas 13083-872, SP, Brazil
6   Laboratory of Emerging Viruses (LEVE), Department of Genetics, Microbiology and Immunology, Institute of Biology, University of Campinas (UNICAMP), Campinas 13083-872, SP, Brazil
7   Experimental Medicine Research Cluster (EMRC), University of Campinas (UNICAMP), Campinas 13083-872, SP, Brazil
8   Laboratory of Aging Biology, Department of Biochemistry and Tissue Biology, University of Campinas (UNICAMP), Campinas 13083-872, SP, Brazil
9   Obesity and Comorbidities Research Center (OCRC), University of Campinas (UNICAMP), Campinas 13083-872, SP, Brazil
10  Biodiversity Research Centre, Federal University of Roraima, Boa Vista 69310-000, RR, Brazil
11  Hospital Israelita Albert Einstein, São Paulo 05652-900, SP, Brazil
12  Department of Biochemistry, Federal University of São Paulo, São Paulo 04044-020, SP, Brazil
*   Correspondence: hmsouza@unicamp.br (H.M.-S.); simabuco@gmail.com (F.M.S.)
†   These authors contributed equally to this work.

**Abstract:** The COVID-19 (Coronavirus Disease 2019), caused by the Severe Acute Respiratory Syndrome Coronavirus 2 (SARS-CoV-2), severely affects mainly individuals with pre-existing comorbidities. Here our aim was to correlate the mTOR (mammalian/mechanistic Target of Rapamycin) and autophagy pathways with the disease severity. Through western blotting and RNA analysis, we found increased mTOR signaling and suppression of genes related to autophagy, lysosome, and vesicle fusion in Vero E6 cells infected with SARS-CoV-2 as well as in transcriptomic data mining of bronchoalveolar epithelial cells from severe COVID-19 patients. Immunofluorescence co-localization assays also indicated that SARS-CoV-2 colocalizes within autophagosomes but not with a lysosomal marker. Our findings indicate that SARS-CoV-2 can benefit from compromised autophagic flux and inhibited exocytosis in individuals with chronic hyperactivation of mTOR signaling.

**Keywords:** COVID-19; single-cell analysis; autophagic flux; SARS-CoV-2; mTOR

---

## 1. Introduction

The coronavirus disease (COVID-19), caused by Severe Acute Respiratory Syndrome Coronavirus 2 (SARS-CoV-2), mainly affects people with pre-existing comorbidities, such as hypertension, diabetes, obesity, and heart disease [1]. Increased signaling through the mammalian/mechanistic Target of Rapamycin (mTOR) signaling pathway is a common characteristic in all comorbidities associated with a higher risk of mortality by COVID-19 [2]. In healthy individuals, mTOR signaling is responsible for maintaining a balance between protein synthesis, autophagy, and nutrient usage and storage processes. This balance is crucial for the cell since its dysregulation leads to cancer, obesity, and diabetes [3].

The process of autophagy, from autophagosome formation to lysosomal degradation, depends on the inactivation of mTOR signaling [4]. Autophagy initiation is induced by the inactivation of mTORC1, preventing the phosphorylation of multiple autophagy-related proteins [4]. However, the activation of mTORC1 is required for the reformation of functional lysosomes at the end of the autophagy flux [5]. Moreover, high mTOR and low lysosomal activations correlate with lower exocytosis [5].

Interestingly, an interactomics study identified mTOR signaling as a pathway that is possibly modulated by SARS-CoV-2 infection [6]. Furthermore, recent findings showed that SARS-CoV-2 infected cells have compromised autophagic flux and increased virus replication [7]. This lead us to believe that the balance between mTOR and autophagy could link comorbidities to severe COVID-19 symptoms.

In this report, we analyzed molecular pathways of SARS-CoV-2 infection in Vero E6 cells and COVID-19 patients' samples to evaluate the balance between the mTOR signaling pathway and the process of autophagy and their correlation with the infection.

## 2. Materials and Methods

### 2.1. Study Description

The present unmatched laboratory-based experimental study combined in vitro results obtained after SARS-CoV-2 infection in Vero cells with in silico re-analysis results from severe COVID-19 patients to understand the role of mTOR and autophagy for viral replication and disease severity.

### 2.2. Vero E6 Cell Culture

Vero E6 (African green monkey, *Cercopithicus aethiops*, kidney) cells were cultured in Dulbecco's modified Eagle medium (DMEM; Sigma-Aldrich, San Diego, CA, USA) supplemented with 10% heat-inactivated fetal bovine serum (FBS; Sigma-Aldrich, San Diego, CA, USA) and 1% Penicillin-Streptomycin 100 U/mL and 100 µg/mL (Sigma-Aldrich, San Diego, CA, USA), and incubated in 5% carbon dioxide atmosphere at 37 °C.

### 2.3. Viral Infection

An aliquot of SARS-CoV-2 SP02.2020 (GenBank accession number MT126808) isolate was kindly donated by Professor Edison Luiz Durigon of the University of São Paulo, São Paulo [8]. Vero E6 cells were used for virus propagation in the Biosafety Level 3 Laboratory (BSL-3) of the Laboratory of Emerging Viruses. Viral infections were performed in Vero cells seeded in 24 wells plates ($5 \times 10^5$ cells/well) for the experiments with treatments and immunofluorescence assays and 6-well plates ($1 \times 10^6$ cells/well) for Western blots. A multiplicity of infection (MOI) of 1 was used for all experiments.

### 2.4. Immunofluorescence

Cells were prepared onto salinized glass slides, fixed, and stained as previously described in the FISH process. Briefly, after fixation with 4% PFA, cells were washed with 0.1 M PBST pH 7.4. After, cells were incubated for 10 min with 0.1 M glycine and treated with BSA solution (Sigma) for 30 min. Cells were then incubated overnight at 4 °C with primary antibodies at 1:100 dilution in PBST and 1% BSA [SARS-CoV-2 Spike S1 antibody (#HC2001 GenScript—#A02038), p62 antibody (#BD 610832), Lamp1 antibody (#BD 555798)

and CD63 antibody (#BD 556019), according to the desired double IF. The slides were washed and incubated for 2 h with secondary antibodies (Alexa 488 anti-Human IgG Thermo Fisher—#A11013 and Alexa Fluor 555 Anti-Mouse IgG #A21422), diluted 1:500 in PBST + 1% BSA. Cells were then washed and stained with DAPI (Santa Cruz Biotechnology, #SC3598). Subsequently, cells were washed with PBS, and the labels were mounted in an aqueous mounting solution for confocal imaging.

### 2.5. Confocal Microscopy

Microscopic images were acquired with an Airyscan Zeiss LSM880 on an Axio Observer 7 inverted microscope (Carl Zeiss AG, Germany) with a C Plan Apochromat 63 × 1.4 Oil DIC objective, 4× optical zoom. Before image analysis, raw.czi files were automatically processed into deconvoluted Airyscan images using Zen Black 2.3 software. DAPI images were acquired as conventional confocal images using a 405 nm laser line for excitation and pinhole set to 1 AU.

### 2.6. VeroE6 RNAseq

Raw RNA sequencing (RNAseq) reads of VeroE6 cells infected with SARS-CoV-2 at a multiplicity of infection (MOI) of 0.3 for 24 h, and mock-infected cells were obtained from SRA (GSE153940) [9,10]. Gene level quantification was performed using Salmon [11] with the *Chlorocebus sabaeus* reference transcriptome obtained from NCBI as the index. Salmon quant mode was run using standard settings and the following flags: –validate mappings and –numBootrstraps 100. The R package tximport [12] was used to load quant.sf files to R and to create a DESeq2 object for differential expression analysis. DESeq2 [13] was used to find differentially expressed genes between SARS-CoV-2 and mock-infected cells using standard settings. Geneset enrichment analysis (GSEA) of differentially expressed genes was performed using the fgsea R package [14] against a custom geneset containing all pathways and terms from Reactome, Gene Ontology (GO) biological process, GO cellular compartment, GO molecular function, Biocarta, KEGG, Hallmark pathways, and Wikipathways. All original GMT files were obtained from the GSEA-MsigDB webservice [15,16]. From the enrichment results containing all terms, pathways related to mTOR signaling, autophagy, lysosome, exocytosis, endosome, late endosome, SNAREs, and vesicle-associated proteins were selected for visualization. Only terms with an adjusted $p$-value < 0.05 were considered for discussion. Leading-edge genes from the significant terms were used to depict results at the gene level. Figures were elaborated using the ggplot2 and pheatmap R packages.

### 2.7. Single-Cell RNAseq (scRNAseq) of Severe COVID-19 Patients

Single-cell transcriptomic data was analyzed from bronchoalveolar lavage fluid (BALF) from patients with varying severity of COVID-19 and matching healthy controls [17]. Clinical and demographic data regarding the patients used in this study can be obtained in the original reference [17]. As described in the original study, patients with severe infection were diagnosed based on one of the following criteria: respiratory distress with respiratory rate $\geq$ 30 times min$-1$; fingertip oxygen saturation $\leq$ 93% at resting state; the ratio of partial pressure of arterial oxygen to fraction of inspired oxygen ($PaO_2/FiO_2$) $\leq$ 300 mm Hg (1 mm Hg = 0.133 kPa); and obvious progression of lesions in 24–48 h shown by pulmonary imaging > 50%. The dataset generated by the original authors is publicly available at https://covid19-balf.cells.ucsc.edu/ (accessed on 12 January 2021). Briefly, the dataset was downloaded, and the RDS seurat object was imported into R environment version v3.6.3. Epithelial cells were selected for downstream analysis based on the cell type annotation provided by the authors. According to the original author's annotation, cells containing viral reads were labeled as "infected," and those in which no viral reads were detected were labeled as "bystanders." Only cells from severe COVID-19 patients presented viral reads; therefore, cells from moderate patients were discarded. Differential expression analysis was conducted using the FindMarkers function

of the seurat package v3.1 [18] using the Wilcoxon test to compare infected and bystander cells of severe patients to healthy control cells and also infected to bystander cells of severe patients. Differentially expressed genes were identified considering genes expressed in at least 10% of cells, FDR < 0.05 and |avg_logFC| > 0.25, according to the standards used by the authors of the Seurat package [18]. Differentially expressed genes found in each comparison were submitted to GSEA using the fgsea R package with the same custom geneset file and selected terms used for the VeroE6 analysis described above. The leading-edge genes of the enriched pathways were selected for plotting differential expression values in a heatmap using the R package pheatmap. Leading-edge genes correspond to the top up- or downregulated genes in the differential expression analysis that are also members of the enriched Reactome pathways. The leading-edge subset can be interpreted as the core of a gene set that accounts for the enrichment signal. Violin plots of the expression of selected differentially expressed genes were produced using the VlnPlot function of the seurat package.

## 2.8. Western Blotting

Proteins were separated by SDS-PAGE and transferred onto nitrocellulose membranes. Nitrocellulose membranes were blocked in a solution of TBS containing 5% nonfat dry milk and 0.1% Tween-20 for 2 h with constant agitation. After blocking, the membranes were incubated with anti-mTOR (Cell Signaling, #2972), anti-p-mTOR (Cell Signaling, #2971), p-S6K1 (Cell Signaling, #9234), anti-S6K1 (Cell Signaling, #2708), anti-pS6 (Cell Signaling, #2215), anti-S6 (Cell Signaling, #2317), anti-LC3 (Cell Signaling, #4108), anti-p62 (Cell Signaling, #88588), and anti-β-actin (Cell Signaling, #4967), antibodies overnight at 4 °C. Membranes were washed with TBS-T (3 times for 10 min) and incubated with horseradish peroxidase-conjugated secondary antibodies anti-mouse (Millipore, #AP308P), or anti-rabbit (INVITROGEN, Thermo Scientific, #31460, San Diego, CA, USA according to the primary antibody for 1 h at room temperature with constant agitation. Bands were visualized using the ECL kit (GE Healthcare). Band densitometry was measured using ImageJ software.

## 2.9. Quantification and Statistical Analysis

GraphPad Prism 8.0 and 9.0 were used for statistical analyses of Western blotting and viral load, respectively. The values presented are means and standard deviation (SD). The mean difference was tested by Test-T analysis. $p < 0.05$ were considered significant. For the differential expression analysis of single cells, the Wilcoxon test was used to compare infected and bystander cells of severe patients to healthy control cells and infected to bystander cells of severe patients. Differentially expressed genes were identified considering genes expressed in at least 10% of cells, FDR < 0.05 and |avg_logFC| > 0.25 and $p < 0.05$ were considered significative.

## 3. Results

We analyzed the mTOR and autophagy pathways in SARS-CoV-2-infected Vero E6 cells. SARS-CoV-2 infected cells showed increased levels of mTOR, S6K1, and S6 phosphorylation by over 3-fold 24 hpi (hours post-infection) compared to the baseline levels of mock cells (Figure 1A). The significantly increased mTOR and S6 phosphorylation levels can also be observed when we use the endogenous gene for their quantification (Figure 1A). We demonstrated that the ratio of the autophagosome markers LC3-II to LC3-I and the content of p62/SQSTM1 increase in infected cells (Figure 1A), which is associated with the accumulation of autophagosomes by blockage of the autophagy flux [7].

Using confocal microscopy, immunofluorescence (IF) for p62 and the viral Spike protein revealed that SARS-CoV-2 colocalizes within autophagosomes in infected cells (Figure 1B), but not with LAMP1 staining, a lysosomal marker (Figure 1C). RNAseq data from SARS-CoV-2-infected Vero E6 cells [9] (Table S1) revealed the down-regulation of genes related to the lysosomal activity (Figure S1), such as APT6AP1, CTSC, and TPP1

(Figure S1). These results suggest that SARS-CoV-2 infection in Vero E6 cells induces the accumulation of autophagosome proteins in a cell environment with increased mTOR activity, a condition known to correlate inversely with lysosomal activity [19].

We then asked whether COVID-19 patient samples would also present some of the alterations observed in SARS-CoV-2-infected Vero E6 cells. We analyzed a publicly available single-cell RNAseq (scRNAseq) dataset of epithelial cells derived from bronchoalveolar lavage fluid (BALF) of severe COVID-19 patients [17] (Figure 1D). We found that both infected and bystander (not infected) BALF epithelial cells of severe COVID-19 patients presented upregulation of mTOR signaling genes (Figure 1E), including *SQSTM1* and *CDKN1A* (Figure S2).

Differential gene expression analysis between infected and bystander BALF epithelial cells of severe COVID-19 patients recapitulated the downregulation of genes related to lysosome and vesicle membrane proteins (Figure 1E) observed in Vero E6 cells (Figure S1). VAMP8 was consistently down-regulated in both infected and bystander BALF epithelial cells, with the expression level observed in infected cells from severe COVID-19 patients (Figure S2). *SQSTM1*, the gene encoding the p62 protein, on the other hand, was up-regulated in bystander or infected cells from severe COVID-19 patients compared to healthy controls, with the highest expression detected in infected cells (Figure S2).

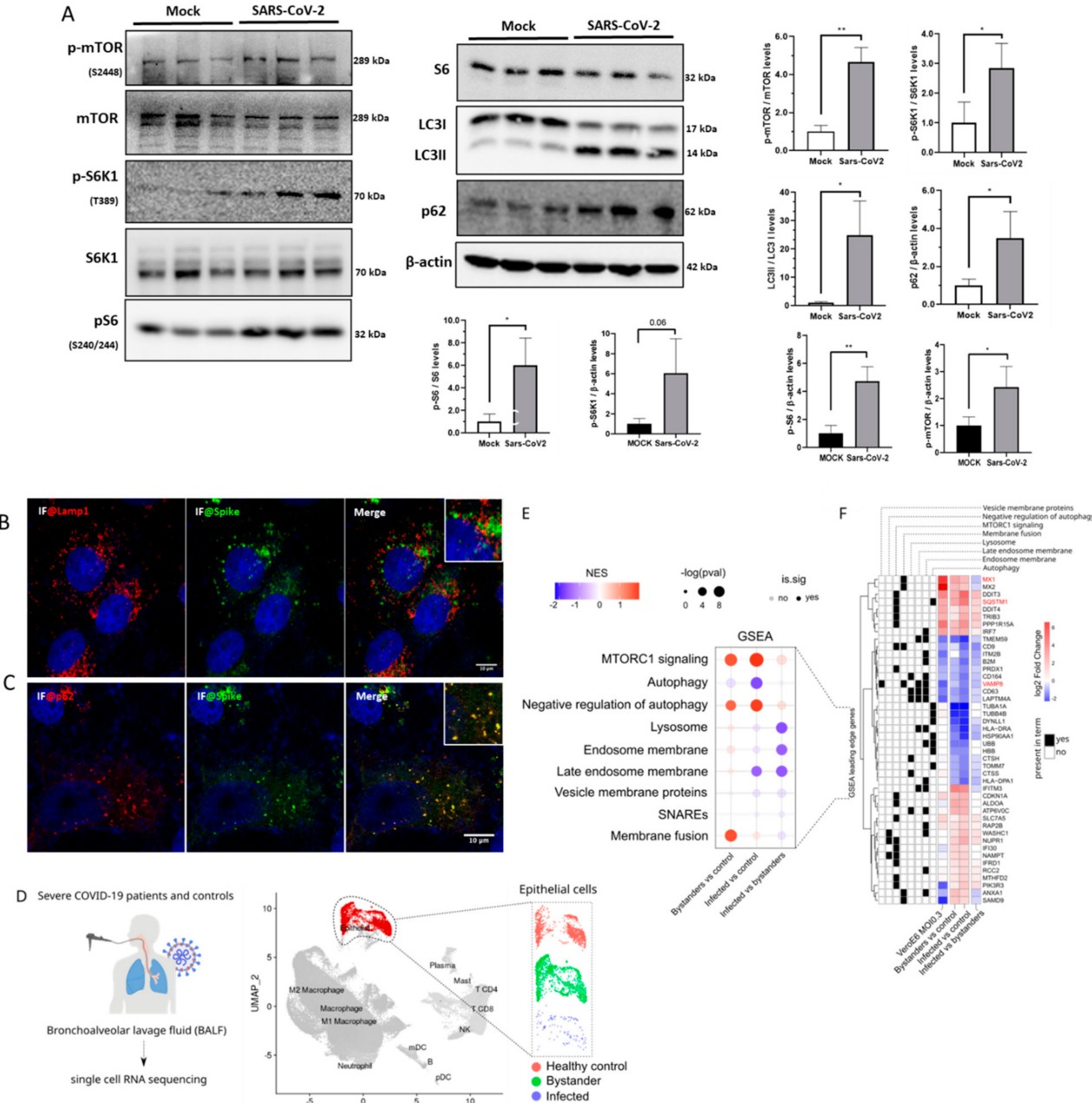

**Figure 1.** SARS-CoV-2 infection activates the mTOR pathway. (**A**) mTOR, p-mTOR, S6K1, p-S6K1, S6, p-S6, LC3I/II, p62, and N immunoblotting of Vero E6 cells protein extracts 24 hpi. β-actin and Vinculin were used as endogenous control (**B**) IF for lysosomal protein LAMP-1 (red) and SARS-CoV-2 Spike protein (green) showing no co-localization (yellow) in Vero E6 cells MOI = 1. (**C**) IF demonstrating co-localization (yellow dots) of autophagosome protein p62 and SARS-CoV-2 Spike protein in Vero E6 cells MOI = 1. Images were acquired with an Airyscan Zeiss LSM880 on an Axio Observer 7 inverted microscope (Carl Zeiss AG, Germany). (**D**) Bronchoalveolar lavage fluid (BALF) samples of severe COVID-19 patients and matched controls were submitted to scRNAseq. Epithelial cells were isolated from the other cell types, and those in which viral reads were detected were labeled as infected. The remaining cells of COVID-19 patients were labeled as bystanders. (**E**) Gene Set enrichment analysis (GSEA) of differentially expressed genes in severe COVID-19 patient bystander, or infected BALF epithelial cells, versus control cells or infected versus bystander

cells of severe patients (Table S1) [17]. (**F**) Fold change values (log2FC) of the leading-edge genes of enriched Reactome terms depicted in E for SARS-CoV-2 infected VeroE6 cells and scRNAseq of severe COVID-19 patients BALF epithelial cells. The leading-edge subset can be interpreted as the core of a gene set that accounts for the enrichment signal. The black and white tiles on the left depict the pathways to the leading-edge genes on the heatmap belong. The data represent means ± SD in samples from independent experiments in each test. *t*-test was used for immunoblotting analysis. * *p* < 0.05 and ** *p* < 0.01 were considered statistically significant.

## 4. Discussion

Though we found activation of the mTOR signaling in SARS-CoV-2 infected cells, our single-cell analysis of severe COVID-19 patients revealed that mTOR signaling is not altered between infected and bystander cells. However, this effect is detected by comparing bystander or infected cells to healthy controls. The same is observed for autophagy-related genes, reduced between COVID-19 patient cells and healthy controls, but not between infected and bystander patient cells. These results suggest that high mTOR signaling and compromised autophagy flux could be a cell response to the viral infection, including those not infected. In addition to blocking autophagy flux, high mTOR activity could favor cap-dependent translation of SARS-CoV-2 genomic and 3′-co-lateral capped subgenomic mRNAs [20]. Since we only observed an increased expression of mTORC1 signaling genes in bystander or infected cells relative to controls but not in infected versus bystander cells, we speculate that this effect could reflect a pre-existing condition of high mTOR signaling and compromised autophagy flux in severe COVID-19 patients.Besides, mTOR activation is important for immune system activation and is related to paracrine signaling [21].

In agreement with a possible causal role of high mTOR signaling favoring COVID-19 severity, mTOR inhibition resulted in a significant reduction in virus replication and may also inhibit viral particle uptake in Huh7, Vero E6 cells, and mucociliary primary human airway-derived air-liquid interface cultures [22]. It is known that hyperactivation of mTOR signaling is present in comorbidities considered most prevalent in deceased COVID-19 patients and advanced-age patients [23]. The inhibition of the mTOR pathway by mTORC1 inhibitors, such as rapamycin and rapalogs, reduced SARS-CoV-2 infection and N protein expression in Vero E6 and primary human airway-derived air-liquid interface cultures, endorsing our results [22].

Another important aspect of our findings is the association between SARS-CoV-2 and autophagosomes. We demonstrated that SARS-CoV-2 colocalizes within autophagosomes in infected cells but not with Lamp1. These results corroborate previous evidence that autophagosome-lysosome fusion is blocked during SARS-CoV-2 infection [24]. Several other groups have shown that coronaviruses can increase autophagosome accumulation and block their fusion with lysosomes through direct interactions between viral and cellular proteins [24,25].

A recent multi-omics study [26] revealed that SARS-CoV-2 infection induces phosphorylation of key proteins in the mTOR pathway, including SQSTM1. The study also shows that SARS-CoV-2 infection promotes the ubiquitination of autophagy-related proteins, including VAMP8, suggesting protein degradation via the proteasome. The results of this multi-omics study agree with our findings, showing that SARS-CoV-2 regulates mTOR activity and autophagy at different levels. Nonetheless, another study showed that SQSTM1/p62 levels did not change during SARS-CoV-2 infection [27]. We demonstrated that the ratio of the autophagosome markers LC3-II to LC3-I and the content of p62/SQSTM1 increase in infected cells (Figure 1A), which is associated with the accumulation of autophagosomes by blockage of the autophagy flux [7].

One of the limitations of the current study is the use of a non-human cell line for analysis of SARS-CoV-2 infection. We understand that follow-ups of our study are needed to further extrapolate our data to other SARS-CoV-2 models. Importantly, however, the Vero E6 cell line has been shown previously to be highly permissible for SARS-CoV-2

infection [28], and previous publications used this cell line as a model for discoveries about the molecular mechanisms of SARS-CoV-2 infection [6,29].

Based on the findings of this study and the data already published in the literature, we present a hypothetical model in Figure 2. Conditions of low mTOR signaling, where autophagy is promoted, and vesicles containing SARS-CoV-2 particles might fuse to acidified lysosomes and be eliminated or go directly to exocytosis, thus exposing the virus to the immune system (Figure 2A). The high mTOR activity could facilitate SARS-CoV-2 infection, ensuring cap-dependent translation of viral proteins and lysosome deacidification and allowing viral assembly in autophagosomes and late endosomes (Figure 2B).

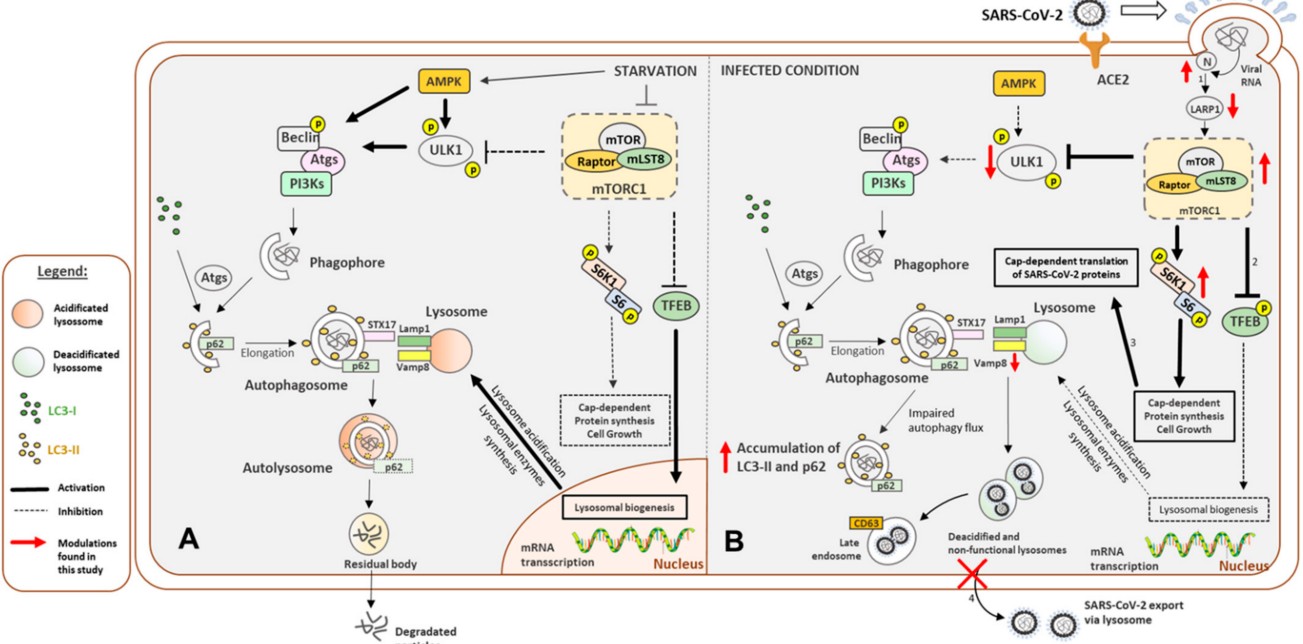

**Figure 2.** Proposed molecular mechanisms of SARS-CoV-2 infection upon high mTOR activation and compromised autophagy flux and vesicle fusion. (**A**) Classical autophagy flux in low mTOR conditions, where SNARE proteins allow for vesicle fusion to the acidified lysosome and vesicle fusion to the cell membrane. (**B**) High mTOR scenario depicting a SARS-CoV-2 viral infection. Newly assembled viral particles associated with cellular vesicles (1) are prevented from being destroyed by lysosomal fusion as the expression of genes coding for vesicle fusion proteins (2), and lysosomal genes (3) are down-regulated, lysosomes are deacidified, and lysosomal fusion to virus-bearing vesicles is prevented. The suppression of vesicle-membrane fusion proteins, such as VAMP8 (4), would prevent virus release from infected cells. Created with BioRender.com.

## 5. Conclusions

Finally, our findings reveal that SARS-CoV-2 infection in Vero E6 cells and severe COVID-19 patients occur in conditions of increased mTOR activity and impaired autophagic flux that could influence viral replication. These data shed light on the role of mTOR and autophagy in SARS-CoV-2 and could be used for therapeutic strategies.

**Supplementary Materials:** The following supporting information can be downloaded at: https://www.mdpi.com/article/10.3390/cimb45010023/s1, Figure S1: Down-regulation of genes related to lysosomes in Vero E6 cells. Figure S2: scRNAseq analysis in COVID-19 patients. Table S1: Differentially expressed genes from scRNAseq analysis in COVID-19 patients and Vero E6 cells infected with SARS-CoV-2.

**Author Contributions:** G.C.M., T.L.D., B.B., M.R.A. and É.P.Z. led and performed experiments, literature searches, data plotting, and analyses. L.N.S., L.B., J.G.d.A.E. and G.C.M. performed IF and HIS, M.C.S.M., I.C.B.P., A.P.M., É.P.Z. and L.G.S.d.S. performed WB, D.A.T.-T., K.B.-d.-S., P.L.P.,

T.D.S. and F.G. performed all SARS-CoV-2 infections in Vero E6 cells and PFU. Í.M.S.d.C. performed bioinformatics. T.L.D., G.C.M., M.R.A. and H.M.-S. generated all figures. G.C.M., M.C.S.M., I.C.B.P., F.M.S. and H.M.-S. conceived the illustrative scheme. G.C.M. and H.M.-S. formatted the manuscript. H.M.-S., F.M.S., J.L.P.-M., H.I.N. and M.A.M. supervised the study, interpreted the results, and contributed to the discussion. H.M.-S. and F.M.S. conceived and designed the study and wrote the manuscript. All the authors revised the manuscript and approved the final version for publication. All authors have read and agreed to the published version of the manuscript.

**Funding:** This work was supported by grants from FAEPEX-UNICAMP (2005/20; 2319/20; 2432/20; 2274/20), São Paulo Research Foundation (FAPESP) (2018/14933-2; 2018/14818-9; 2020/05284-0; 2014/50938-8; 2020/05346-6; 2020/04919-2; 2020/04558-0) and National Council for Scientific and Technological Development (CNPq) (465699/2014-6).

**Data Availability Statement:** The data presented in this study are available on request from the corresponding author.

**Acknowledgments:** The authors acknowledge the National Institute of Science and Technology of Photonics Applied to Cell Biology (INFABIC) and the technical support of Mariana Ozello Baratti for her aid with confocal microscopy and Elzira Saviani and Thais Theizen for their technical support. Hernandes F. Carvalho and Luis Lamberti Pinto da Silva for providing important primary and secondary antibodies related to markers of cellular and biological processes, and Guilherme Barbosa, Raissa G. Ludwig, and Thiago L. Knittel for thorough scientific discussions and critical reading of the manuscript.

**Conflicts of Interest:** The authors declare no conflict of interest.

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
