# Peer review of "Increased mTOR Signaling and Impaired Autophagic Flux Are Hallmarks of SARS-CoV-2 Infection"

_cimb, doi:10.3390/cimb45010023_

Round 1

Reviewer 1 Report

It is a well-designed study useful for future of SARS-COV-2 patients management and for understanding of virus behavior.

Author Response

Thank you for the careful revision of our manuscript. 

Reviewer 2 Report

In this manuscript the authors insisted  that SARS-CoV-2 infection in Vero E6 cells and severe COVID-19 patients occur in conditions of increased mTOR activity and impaired autophagic flux that could influence viral replication. These data shed light on the role of mTOR and autophagy in SARS-CoV-2 and could be used for therapeutic strategies.

I have some comments.

1. In the figure 1E and the discussion, the authors insisted mTOR signaling is not altered between infected and bystander cells.I understand that the SARS-CoV-2 virus activates mTOR, but please tell me the mechanism bystander cell mTOR activation.

2. Is mTOR activation by viral infection a characteristic phenomenon of CoV-2? Or is it a common phenomenon?

3. Is it common to give rapamycin or rapalog to people with viral infections?

4. Does suppression of mTOR in bystander cells make infection less likely to occur?

Reviewer 3 Report

I would like to thank the handling editor for giving me the opportunity to review the paper entitled “Increased mTOR signalling and impaired autophagic flux are hallmarks of SARS-CoV-2 infection” by Zambalde and colleagues. This is an experimental study investigating molecular pathways of SARS-CoV-2 infection in Vero E6 cells and samples from patients with COVID-19 to assess the balance between the mTOR signalling pathway and the process of autophagy, as well as their correlation with SARS-CoV-2 infection. The authors found increased mTOR signalling and suppression of genes associated with autophagy, lysosome, and vesicle fusion in Vero E6 cells infected with SARS-CoV-2, as well as in transcriptomic data mining of bronchoalveolar epithelial cells from patients with severe COVID-19. Furthermore, they performed co-localization assays which indicated that SARS-CoV-2 colocalizes within autophagosomes but not with lysosomal marker. Therefore, the authors conclude that SARS-CoV-2 can benefit from compromised autophagic flux and inhibited exocytosis in individuals with pre-existing comorbidities and chronic hyperactivation of mTOR signalling.

This is an interesting and informative study that has the potential to be a valuable addition to the pertinent literature. The introduction sets the appropriate background even for the reader with little knowledge on the topic. The methods are adequately described to allow replication of results, which are clearly presented. Finally, the conclusions are supported by the findings of the study. Here, I have made a few suggestions that (in my opinion) could help improve the overall quality of the manuscript.

·         In the Abstract, the authors may consider clearly stating the objective of the study and further detailing the methodology.

·         In the Materials and Methods, the authors may consider explicitly stating the specific design of their study (e.g., matched/unmatched laboratory‐based experimental study).

·         The authors may consider further elaborating on the statistical analyses performed, as more tests were used (e.g., Wilcoxon test) than those reported in the relevant section of the methodology.

·         The authors may consider clarifying how the severity of infection (e.g., moderate versus severe) was defined in the patients with COVID-19 who were included in the study, as well as how these patients were selected.

·         The authors may consider presenting demographic and clinical characteristics (e.g., comorbidities) of the patients with COVID-19 who were included in the study.

·         The authors may consider discussing limitations of their study.

Round 2

Reviewer 3 Report

Thank you for considering my suggestions and revising your manuscript accordingly.